# A high-yield protein expression platform in the unicellular red alga *Cyanidioschyzon merolae*

Yuko Mogi[1], Shogo Tsushima[1], Shotaro Nagai[1], Shinichi Gima[2], Fumi Yagisawa[3] and Yamato Yoshida[1,4,*]

## ABSTRACT

The production of engineered proteins in transgenic cells is widely used in research, medicine and industry. However, conventional cell-based production systems still face challenges in cost, scalability and biosafety. Here, we present a recombinant protein expression platform with simplified purification based on the photosynthetic unicellular red alga *Cyanidioschyzon merolae*, which can be cultivated under highly acidic conditions using only inorganic nutrients, air, water and light. We first identified a promoter that drives high-level constitutive gene expression throughout the cell cycle, resulting in substantial mRNA accumulation in *C. merolae*. A stable transformant expressing His-tagged mVenus under the control of this promoter accumulated the recombinant protein to more than 1% of total soluble protein. The simple cellular architecture of *C. merolae*, including the absence of a cell wall, enables efficient protein extraction via a single freeze–thaw cycle, followed by purification using immobilized metal affinity chromatography (IMAC), yielding ~13.9 mg of functional recombinant protein per gram of total soluble protein. Owing to its low cost, scalability, operational simplicity and minimal risk of contamination, this *Cyanidioschyzon*-based platform offers a practical and promising approach to recombinant protein production in a photosynthetic eukaryote.

KEY WORDS: Recombinant protein purification, Alga, *Cyanidioschyzon merolae*

## INTRODUCTION

Proteins play fundamental roles in cellular processes owing to their diverse functions. To elucidate the function and molecular mechanisms of each protein, obtaining a sufficient quantity and quality of the target protein is essential for conducting comprehensive analyses. Furthermore, purified proteins, such as antibodies and enzymes, are widely used in medical and industrial applications, further underscoring their importance and necessity. To address these demands, numerous protein purification systems have been developed to date. In addition to isolating natural proteins from living tissues, advancements in genetic recombination, transformation and molecular biology technologies have enabled the generation of cells that express desired proteins or the synthesis of proteins of interest through cell-free protein synthesis systems (Nishikawa et al., 2001; Wingfield, 2015). However, despite their practicality and convenience, these production systems still face challenges and limitations, particularly in terms of cost, scalability and product safety. Contamination risks, such as endotoxins derived from lipopolysaccharides in the membranes of gram-negative bacteria and viral contamination in cultured cells, necessitate rigorous monitoring and exclusion procedures (Barone et al., 2020; Wingfield, 2015). Furthermore, the use of highly enriched medium in many systems hinders efforts to reduce the production costs of purified proteins.

In recent decades, microalgae have gained attention as a novel platform for recombinant protein production (Banerjee and Ward, 2022). Algae grow through photosynthesis, relying solely on simple resources such as water, carbon dioxide and light. Therefore, their cultivation can be achieved at comparatively low cost, making them a potentially cost-effective alternative to conventional production systems. For example, green algae such as *Chlamydomonas reinhardtii* and *Chlorella* species are commonly used due to their relatively simple cultivation requirements (Rasala and Mayfield, 2015; Yang et al., 2016). Established transformation techniques allow for the stable introduction of target genes encoding recombinant proteins into these host cells. The transformed algal strains are then grown in controlled cultivation systems, such as bioreactors, followed by protein purification through extraction and chromatographic methods. These algae can be cultivated using renewable resources, like sunlight and carbon dioxide, and scaled up in bioreactors to produce recombinant proteins.

Nevertheless, the expression levels of recombinant proteins in algal systems generally remain significantly lower than those achieved in traditional hosts such as *Escherichia coli*, yeast or mammalian cells (Banerjee and Ward, 2022). For nuclear expression in *C. reinhardtii*, recombinant protein yields typically range from 0.01 to 0.1% of total soluble protein (TSP), corresponding to ~3–30 µg per gram of wet cell mass. Although the expected yield increases to ~0.5–5% of TSP in the case of chloroplast expression (Ma et al., 2022; Taunt et al., 2018), the lack of glycosylation capability limits the functional properties of recombinant proteins, and the regulatory flexibility of gene expression in the *Chlamydomonas* chloroplast remains restricted. Thus, although algal systems offer ecological and economic advantages, their low productivity and inherent limitations in genetic engineering underscore the need for alternative platforms that combine high-level expression with molecular flexibility and robust protein yields.

Among various microalgae, we focused on the unicellular red alga *Cyanidioschyzon merolae* as a promising platform for recombinant protein production. *C. merolae* cells possess a minimal set of membrane-bound organelles, and the absence of a cell wall allows for efficient isolation of intracellular components, such as organelles and endogenous molecules (Kuroiwa, 1998; Matsuzaki et al., 2004). Furthermore, because *C. merolae* thrives under highly acidic

[1]Department of Biological Sciences, Graduate School of Science, The University of Tokyo, Tokyo 113-0033, Japan. [2]Integrated Technology Center, University of the Ryukyus, Okinawa 903-0213, Japan. [3]Research Facility Center, University of the Ryukyus, Okinawa 903-0213, Japan. [4]Japan Science and Technology Agency (JST), FOREST, Saitama 332-0012, Japan.

*Author for correspondence (yamato.yoshida@bs.s.u-tokyo.ac.jp)

Y.M., 0009-0002-4584-3047; F.Y., 0000-0002-8534-3824; Y.Y., 0000-0002-2339-2938

(pH 2–3) and high-temperature (∼40–45°C) conditions, very few other organisms can grow in the same environment. As a result, *C. merolae* can be selectively cultivated without the need for antimicrobial agents or other selective chemicals. *C. merolae* carries out simple *N*-linked glycosylation, which might support recombinant protein folding and solubility through stabilizing glycan–protein interactions (Hebert et al., 2014; Mathieu-Rivet et al., 2014; Schulze et al., 2020). In addition, the genome of *C. merolae* has been fully sequenced, and several genetic manipulation techniques, including homologous recombination and CRISPR-based methods, have been established for this organism (Fujiwara et al., 2015, 2017; Imamura et al., 2009; Ohnuma et al., 2008; Tanaka et al., 2021).

In this study, we present a highly efficient protein expression and purification platform using *C. merolae* cells. A newly identified promoter and engineered cloning plasmids containing this promoter enable stable and exceptionally high levels of transgene expression in *C. merolae*. Combined with a simple freeze–thaw lysis procedure and Ni$^{2+}$-nitrilotriacetic acid (Ni-NTA) affinity chromatography, our system enabled purification of up to ∼13.9 mg of target protein per gram TSP. We further demonstrate the purification of mVenus and an anti-GFP-VHH (anti-GFP nanobody) using this platform, including an immunoprecipitation-based purification.

Thus, the *Cyanidioschyzon*-based platform, together with a high-level expression promoter, provides a versatile and highly efficient approach for recombinant protein production and purification in a photosynthetic eukaryote, offering an innovative and potentially low-cost strategy for generating purified proteins.

## RESULTS

### Identification of the optimal gene promoter for achieving high-level and stable protein expression in *C. merolae*

To establish the *Cyanidioschyzon*-based protein purification platform, we first sought to identify a gene promoter capable of driving high and stable levels of protein expression in *C. merolae*. By comparing mRNA copy numbers and ribosome footprint counts from transcriptome and translatome datasets of *C. merolae* cells in both non-dividing and dividing phases (Mogi et al., 2025), we identified a gene, distinguished by gene ID *CMK024C*, that consistently exhibited the highest expression throughout the cell cycle, in terms of both transcription and translation (Fig. 1A). The *CMK024C* gene, designated as *HiX* (for 'high-level expression'), encodes an uncharacterized protein consisting of 172 amino acids. Although the functional role of the HiX protein remains unknown, its promoter drives exceptionally high expression levels, exceeding those of classical high-expression genes, such as *UBC*, which encodes ubiquitin C, suggesting that *HiX* might function as a housekeeping gene and that its promoter is highly suitable for use in the *Cyanidioschyzon*-based protein purification system (Fig. 1B). Deep sequencing analysis also predicted a putative transcription start site (TSS) located ∼165 base pairs upstream of the first methionine codon (Fig. 1C, arrow). Notably, its upstream region is enriched with cytosine-phosphate-guanine (CpG) motifs (Fig. 1D), which are genomic features typically associated with housekeeping genes and TATA-less promoters (Gardiner-Garden and Frommer, 1987). A 500-bp segment between positions −550 and −50 within the 1000-bp upstream sequence exhibits a GC content of 56.4% and a CpG observed/expected ratio of 1.36 (Fig. 1E,F), fulfilling the classical criteria for a CpG island (Gardiner-Garden and Frommer, 1987).

To gain an initial insight into the function of HiX driven by the high-expression promoter, we predicted its protein structure and examined its intracellular distribution. The *HiX* gene product was predicted to adopt a bundle structure comprising four α-helices (Fig. 1G,H). mVenus-tagged HiX localized to the cellular membrane, which is compatible with the predicted four-helix bundle structure, suggesting that HiX might function as a membrane-associated protein *in vivo* (Fig. 1I).

Based on these results, we conclude that the 1000-bp upstream region from the first methionine is sufficient to function as a promoter sequence for gene expression regulation. Hereafter, the 1-kb upstream sequence of *HiX* is referred to as the high-level expression promoter (HiX promoter) and is used as a strong constitutive promoter in the *Cyanidioschyzon*-based protein purification system.

### Design of an expression vector for the *Cyanidioschyzon*-based protein purification system

We subsequently constructed an expression vector driven by the HiX promoter to achieve high-level production of the target protein in *C. merolae* cells. The engineered vector, pHiX, contains the HiX promoter and a gene cassette for the *URA5.3* selection marker (Fig. 2A). These sequence elements are flanked by homologous regions associated with a genetic safe harbor locus on both sides. By embedding a gene encoding the target protein into the cloning site, its expression is driven by the HiX promoter. DNA fragments amplified by PCR using the pHiX vector can be introduced into *C. merolae* uracil-auxotrophic M4 cells, and the resulting transformants are expected to produce the target protein, which is constitutively expressed and accumulates in the cytoplasm (Fig. 2B).

### Cellular-level evaluation of transgene expression driven by the strongest *C. merolae* promoter

To evaluate the stability and expression level of a transgene *in vivo*, we examined the transformation of a gene encoding the yellow fluorescent protein mVenus using the genetic scheme. As a result, no delays or differences in growth rates were observed between the transformants and wild-type cells. Fluorescence microscopy revealed that mVenus protein products were localized in the cytoplasm with very strong fluorescence intensity, and no abnormalities were observed in the transformants (Fig. 2C). Additionally, we found no significant differences in fluorescence intensity between dividing and non-dividing cells, indicating that transgene protein expression is stably maintained throughout the cell cycle.

However, we observed heterogeneity in expression levels across the transformant cell lines. By comparing the fluorescence intensities of 15 strains, we identified four statistically distinct groups based on their expression levels (Fig. 2D). The ratios of mean fluorescence intensity of the moderate-expressing strain (#1) and the highest-expressing strain (#52) relative to the lowest-expressing strain (#23) were ∼3.9 and ∼7.0, respectively. Supporting this, mVenus RNA expression in strains #1 and #52 was 2.0- and 4.8-fold higher, respectively, compared to strain #23, as determined by qPCR (Fig. 2E,F). These data suggest that multi-copy insertions or other unpredictable effects might have influenced expression activity in some strains.

No detectable degradation products were observed in total cellular lysates from transformant cells in both SDS-PAGE followed by Coomassie Brilliant Blue (CBB) staining and immunoblot analysis (Fig. 3A,B). In addition, the abundance of mVenus in strain #23 (low expression), #1 (moderate expression) and #52 (high expression) was ∼1.27%, ∼1.46% and ∼3.20% of the TSP, respectively (Fig. 3C,D).

Given the purpose of target protein purification, establishing strains with moderate expression levels (group c) appears to be the most practical approach, as such strains were obtained in four out of

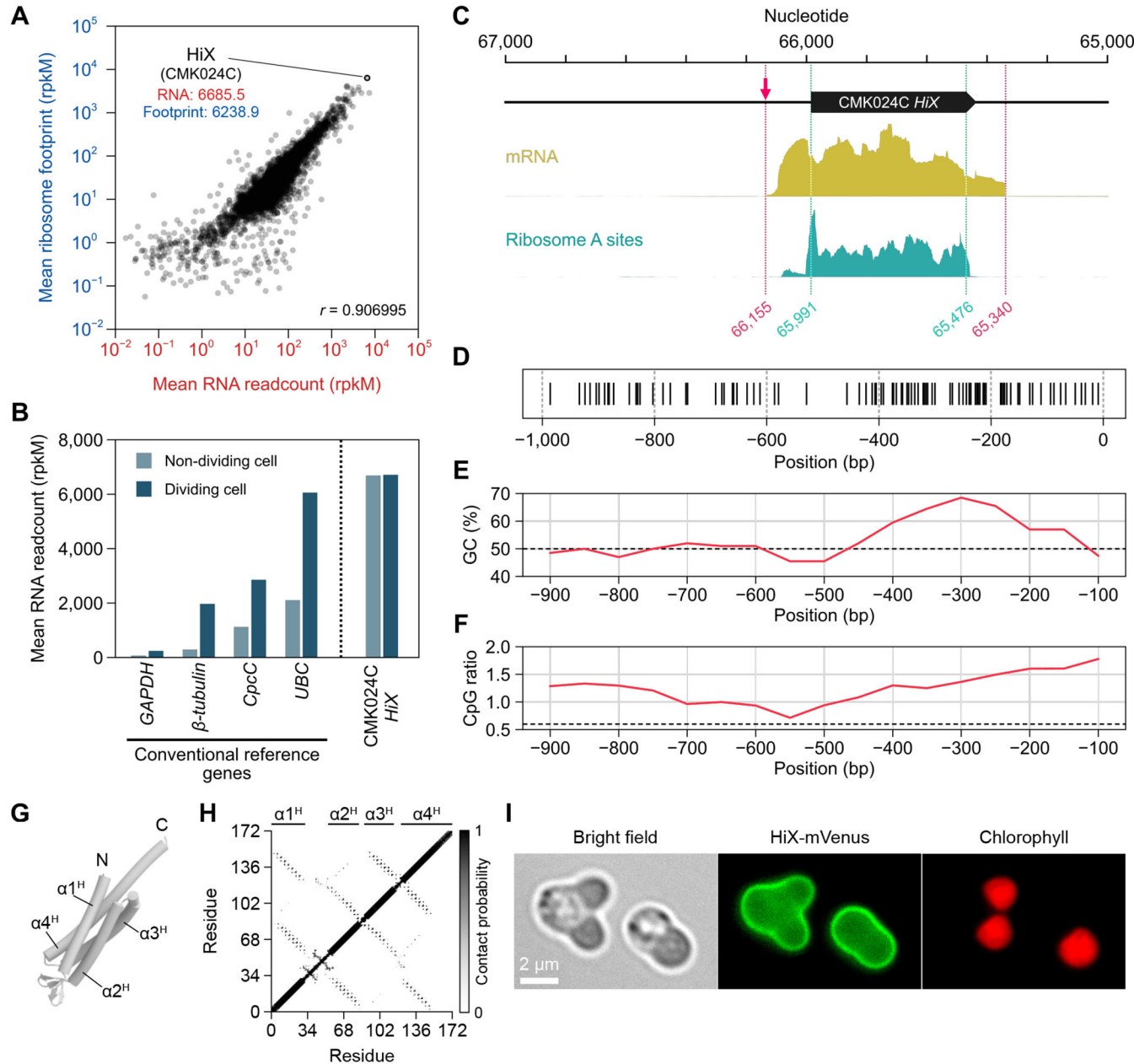

**Fig. 1. Characterization of the gene promoter that exhibits the highest RNA transcription level throughout the cell cycle in *C. merolae*.** (A) Comparison of RNA transcription levels and protein translation levels, measured by RNA-seq and ribosome footprint density, for each nuclear DNA-encoded gene. Ribosome footprint and RNA transcription data were adapted from Mogi et al. (2025) (GEO accession number GSE273424). (B) RNA transcription levels of conventional reference genes and *HiX* (*CMK024C*) in non-dividing (left) and dividing (right) cells. Data in A and B represent the mean values from two independent experiments. (C) mRNA and ribosome footprint densities across the gene locus of *HiX*, including its 5′ and 3′ flanking regions, highlighting the untranslated regions. A putative transcription start site is indicated by an arrow. (D) Distribution of CpG sites within the 1000-bp upstream region of the *HiX* gene. Each CpG site is represented as a black vertical bar. (E,F) GC content (E) and CpG observed/expected ratio (F) across the 5′ flanking region of *HiX*, calculated using a 200-bp sliding window with a 50-bp step size. Dashed lines indicate the 50% GC content threshold (E) and the observed/expected CpG ratio threshold of 0.6 (F), respectively. (G,H) Predicted structure and contact probability map of the HiX protein. The structural model was generated using AlphaFold3 (Abramson et al., 2024) with the 172-amino-acid HiX protein sequence as input. The model predicts that HiX consists of four α-helices, forming interhelical contacts between $\alpha1^H$–$\alpha2^H$, $\alpha2^H$–$\alpha3^H$, $\alpha3^H$–$\alpha4^H$ and $\alpha1^H$–$\alpha4^H$. (I) Cellular expression of HiXp::HiX–mVenus. A non-dividing cell (right) and a dividing cell (left) are shown as representative images. The cell line expressing HiX–mVenus was established by homologous recombination, as described previously (Mogi et al., 2025). Images are representative of more than three independent experiments.

the 15 transformant cell lines (∼26.7%). We therefore selected strain #1, which exhibited a moderate expression level, for subsequent analyses.

Taken together, these results indicate that the transgene expression system using the HiX promoter ensures high reliability in both the yield and quality of the recombinant protein.

**Streamlined purification of recombinant proteins from *C. merolae* cells**

We next established a simple method for isolating target proteins from a transformant cell line expressing His-tagged mVenus fusion proteins (Fig. 4A). Cells were cultured under light for 2 days in 0.7 l of conventional ammonium-based medium with aeration (Fig. 4B,

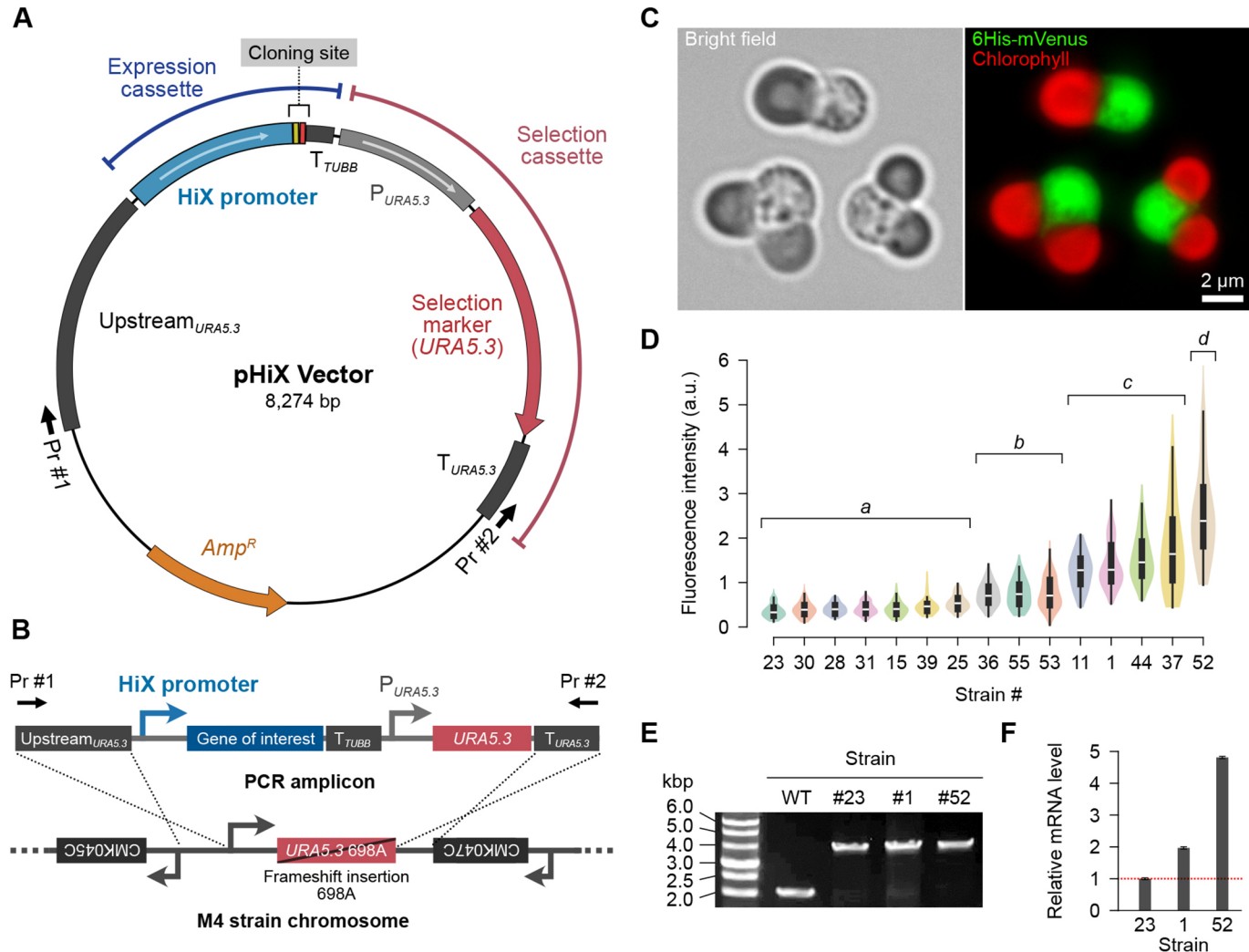

**Fig. 2. Genetic strategy for high-level expression of a transgene in *C. merolae*.** (A) Plasmid map of the pHiX vector DNA. (B) Schematic overview of the transformation strategy using PCR-amplified fragments generated with primers #1 and #2 in *C. merolae* M4 cells. (C) Cellular expression of HiXp::mVenus. A non-dividing cell (top) and two dividing cells (bottom left and right) are shown as representative. Images are representative of more than three independent experiments. (D) Statistical grouping based on fluorescence intensity in each strain. Pairwise comparisons between strains were performed using the two-sided Mann–Whitney–Wilcoxon test with a significance threshold of $P<0.01$. Strains sharing the same letter are not significantly different. Box boundaries represent the 25th and 75th percentiles; horizontal lines within boxes indicate medians; whiskers denote minimum and maximum values. Sample sizes (number of cells per strain) were as follows: #23, 87; #30, 84; #28, 47; #31, 69; #15, 64; #39, 44; #25, 82; #36, 45; #55, 52; #53, 63; #11, 54; #1, 79; #44, 40; #37, 49; #52, 54. (E) DNA electrophoresis confirming the insertion of the DNA cassette at the target locus. (F) Quantitative RT-PCR analysis showing the expression levels of the HiX promoter-driven *mVenus* gene in strains #23, #1, and #52. Data are presented as mean±s.d. ($n$=2). a.u., arbitrary units.

left) and then harvested by gentle centrifugation (2000 *g* for 3 min) (Fig. 4B, middle). After washing with buffer, the cell pellet was rapidly frozen in liquid nitrogen and then thawed in lysis buffer, turning the lysate a deep blue color (Fig. 4B, right). Owing to the unique characteristics of *C. merolae*, which lacks a thick and rigid cell wall, cellular proteins were efficiently released by a single freeze–thaw cycle. Complete cell disruption and release of cellular components were confirmed by microscopy (Fig. 4C). The resultant lysate was filtered and then applied to a nickel–Sepharose column. Using IMAC followed by dialysis (Fig. 4D), a total of 0.54 mg of His-tagged mVenus was purified from 38.9 mg of TSP, corresponding to 13.9 mg per gram TSP. SDS-PAGE followed by CBB staining revealed a single band corresponding to the predicted molecular mass of mVenus, with no apparent contamination (Fig. 4E).

Despite the simplicity of the procedure, the high yield makes this an efficient and practical method for purifying recombinant proteins. Although we demonstrated protein purification from a

0.7 l cell culture, scaling up the culture volume is straightforward. The frozen cell pellet can be conveniently stored at −80°C. The cost of the medium is also highly economical, as it does not require any organic reagents or antibiotics, and *C. merolae* cells can be cultivated in a standard incubator equipped with conventional white LED lighting. Taken together, we propose the *Cyanidioschyzon*-based protein purification system as a simple and high-yielding platform for recombinant protein purification using *C. merolae*.

### Selective protein purification coupled with immunoprecipitation in the *Cyanidioschyzon* protein expression platform

To demonstrate the versatility of the *Cyanidioschyzon*-based protein purification system driven by the HiX promoter, we next performed immunoprecipitation-based purification using mVenus expressed in *C. merolae* and evaluated the enrichment specificity of the workflow. *C. merolae* has been widely used as a model organism

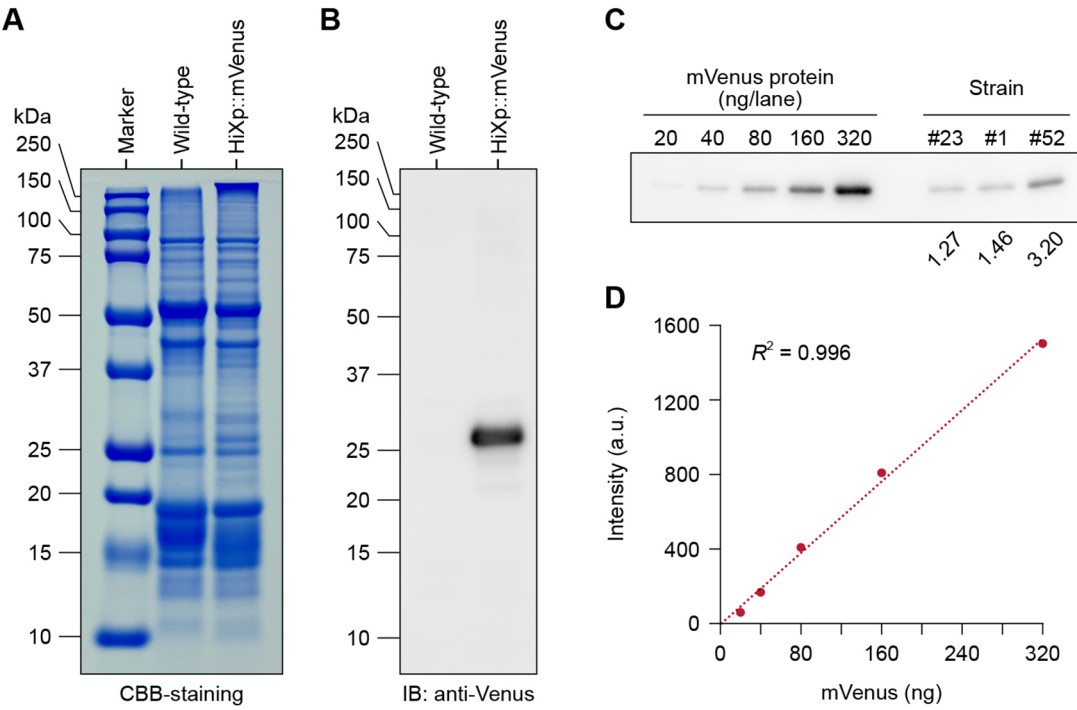

**Fig. 3. Quantification of protein expression levels of the HiX promoter-driven *mVenus* gene.** (A,B) Protein profiles (A) and immunoblot (IB) analysis (B) of wild-type and HiXp::*mVenus*-expressing cells. (C,D) Quantification of total soluble mVenus protein in cells. Based on a calibration curve generated with purified mVenus (D), total soluble mVenus levels in strains #23, #1, and #52 were quantified by immunoblot analysis using an anti-His tag antibody and were calculated as 1.27%, 1.46%, and 3.20% of total soluble protein (*n*=1), respectively. a.u., arbitrary units.

to investigate fundamental principles and molecular mechanisms in cell biology. Therefore, establishing a systematic approach for identifying protein components belonging to a complex in this system will facilitate deeper insights into these cellular processes.

By using an affinity resin for immunoprecipitation, we successfully purified mVenus with a yield of 3.8 mg mVenus per gram of total soluble protein. SDS-PAGE followed by CBB staining revealed a single band corresponding to mVenus (Fig. 5A), and the purity of the fraction was confirmed by mass spectrometry (Fig. 5B). The top score corresponding to mVenus was clearly higher than background signals derived from chloroplast-associated photosynthetic proteins, indicating that the combination of immunoprecipitation and mass spectrometry using this system is suitable for identifying components within protein complexes.

Based on this approach, we next examined the purification of FLAG-tagged anti-GFP-VHH from *C. merolae* cells. A high-expressing transformant cell line (#7) was selected by comparing four transformants using immunoblot analysis (Fig. 5C), and the FLAG-tagged anti-GFP-VHH was purified with a yield of 1.2 mg per gram of TSP. Using the purified anti-GFP-VHH as a standard, the abundance of anti-GFP-VHH in strain #7 was calculated to be 0.34±0.04% TSP (Fig. S1). The purified anti-GFP-VHH also exhibited mVenus-binding activity comparable to that of a conventional IgG anti-GFP antibody, as confirmed by immunoblot analysis (Fig. 5E).

In summary, we successfully purified both a fluorescent reporter (mVenus) and an anti-GFP-VHH using the same platform. Overall, this system consistently enabled efficient purification with multiple affinity tags, supporting broad applicability to diverse target proteins.

## DISCUSSION

In this study, we developed the *Cyanidioschyzon*-based protein purification system as a simple, robust and high-yield platform for

recombinant protein purification using the red alga *C. merolae*. Below is a simplified, experiment-based outline of the recombinant protein purification procedure using the *Cyanidioschyzon*-based system. The transgenesis workflow is fully systematized: gene cassettes are constructed using the pHiX vector via Gibson assembly (Fig. 2A), followed by PCR amplification and introduction into the *C. merolae* M4 strain through a conventional transformation protocol (Fig. 2B). Highly expressing transformant cell lines without abnormal growth phenotypes are then selected by microscopy observation and immunoblot analysis using an appropriate anti-tag antibody. Although transformants can be stored by cryopreservation until use (Ohnuma et al., 2011), they can also be stably maintained in liquid culture, requiring subculturing only once per month. For protein production, ~10% of the culture volume is used as an inoculum and transferred to fresh medium, followed by cultivation for 24–48 h. Harvested cells are rinsed with an appropriate buffer, and the resulting pellets are immediately frozen and stored until use. Upon resuspension and lysis of the frozen pellets, recombinant proteins expressed in the cytosol are efficiently released into the buffer without the need for additional cell disruption steps (Fig. 4C). The target protein can then be purified from the lysate using conventional chromatographic methods, such as IMAC (Fig. 4D), or by immunoprecipitation (Fig. 5A,D).

Despite its simplicity, this method achieves remarkably high yields compared with that seen in other algal recombinant protein production and purification systems (Banerjee and Ward, 2022). In addition to the *Cyanidioschyzon*-based workflow, several microalgal platforms have been reported as potential high-yield systems for recombinant protein production. For example, mVenus has been expressed as a transgene in the microalga *Nannochloropsis oceanica* at levels of ~4.9% of total soluble protein (de Grahl et al., 2020). However, the rigid cell wall of *N. oceanica* necessitates harsh

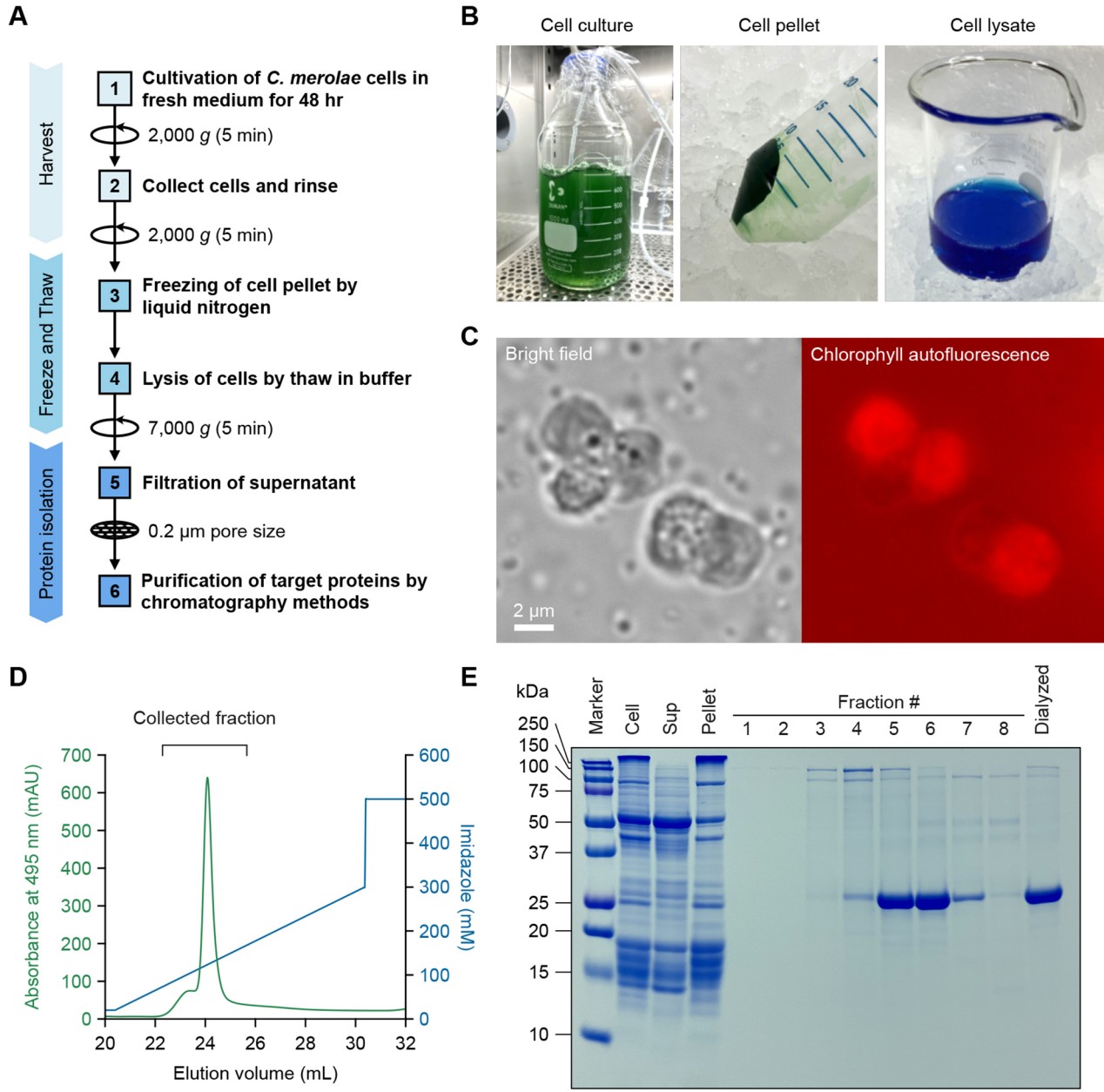

**Fig. 4. Workflow of the protein purification process in the *Cyanidioschyzon*-based protein purification system.** (A) Schematic overview of the protocol for recombinant protein isolation using the *Cyanidioschyzon*-based protein purification system. (B) Representative images showing a 0.7-l culture in a glass bottle (left), harvested cell pellet (middle) and resulting cell lysate (right). (C) Microscopy image of disrupted cells following a single freeze–thaw cycle. Images are representative of more than three independent experiments. (D) IMAC of 6×His-tagged mVenus from the cell lysate. (E) SDS-PAGE analysis of the cell lysate (Cell), supernatant (Sup) and cell debris pellet (Pellet), IMAC elution fractions, and a dialyzed sample from fractions #5 and #6. For the Cell, Sup and Pellet samples, 20 µg of total protein was loaded per lane. For the IMAC elution fractions and the dialyzed sample, 10 µl of each was loaded. Data in D and E representative of three experimental repeats.

cell-disruption procedures to extract recombinant proteins, which can reduce recovery during downstream purification. As a result, the final yield of purified protein is often substantially lower than the apparent expression level. To address the potential loss of recombinant proteins during cell lysis and purification, a secretion-based production platform has also been established in the green alga *Picochlorum renovo*. This species offers unique advantages, including rapid growth (~2 h doubling time) and the genetic tractability of its nuclear genome (Krishnan et al., 2025). It has been reported that, using this system, N-terminal secretory signal peptides fused to the red fluorescent protein mCherry promote

secretion into the culture medium, achieving a production rate of 0.19 mg/l/day (Dahlin and Guarnieri, 2021). This secretion-based strategy is potentially very attractive for algal protein production; however, the secreted target proteins still need to be recovered from the culture medium, typically by IMAC or other appropriate purification methods. Given that *P. renovo* still requires downstream purification from the culture medium, where the risk of contamination can increase, the *Cyanidioschyzon*-based system retains clear advantages.

The *Cyanidioschyzon*-based platform minimizes contamination risks associated with bacterial toxins and viral pathogens. Ammonium-based medium maintains a highly acidic pH of 2.0–2.5, under which

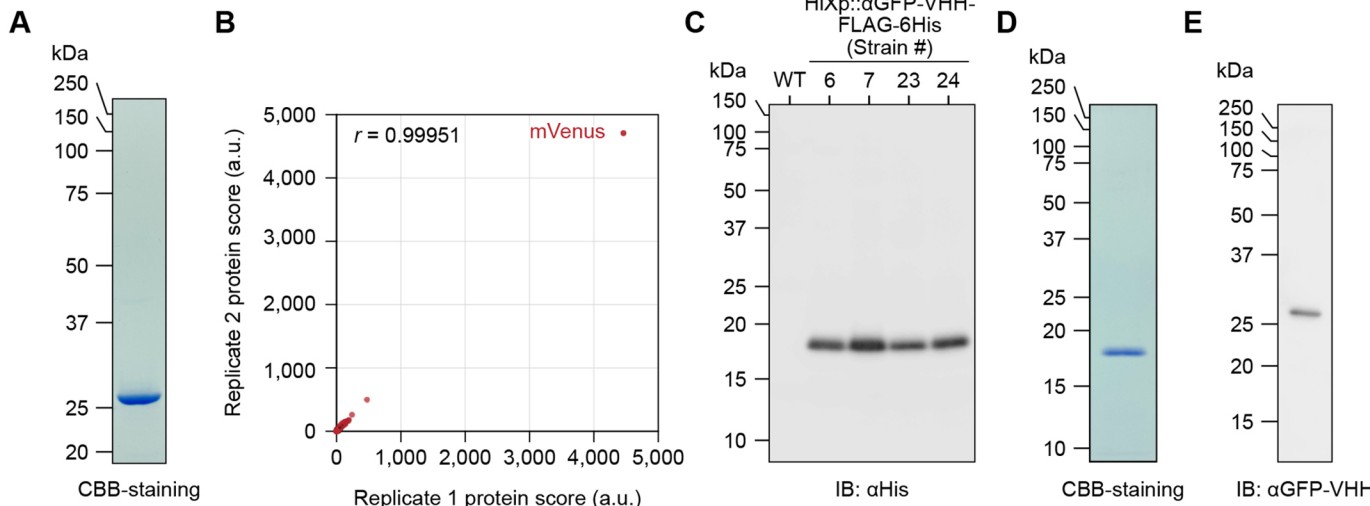

**Fig. 5. Selective protein purification coupled with immunoprecipitation in the *Cyanidioschyzon*-based platform.** (A) SDS-PAGE analysis of purified mVenus protein obtained through the immunoprecipitation approach. Data representative of three experimental repeats. (B) Scatter plot comparing protein scores from LC-MS/MS analysis of the purified mVenus fraction across two technical replicates. The highest-scoring protein (4457.79 in replicate 1 and 4709.16 in replicate 2) was identified as mVenus, with a score ~9.5-fold higher than that of the second-ranked protein. (C) Immunoblot (IB) analysis of wild-type and HiXp::αGFP-VHH-expressing cells. Relative signal intensities of transformant strains #6, #7, #23, and #24 were calculated as 1.06, 1.44, 1.00 and 1.15, respectively, using strain #23 as the reference (set to 1.00). (D) SDS-PAGE analysis of purified anti-GFP-VHH protein obtained through the immunoprecipitation approach. (E) Detection of mVenus protein by immunoblot analysis using purified anti-GFP-VHH. Loaded sample amounts: 3 µg of purified mVenus (A); 20 µg of cell lysate per lane (C); 2 µg of purified anti-GFP-VHH (D); 200 ng of purified mVenus (E). Data in C and D are representative of two independent experiments, and data in E are representative of three independent experiments. a.u., arbitrary units.

most bacteria and human-infectious viruses cannot survive. The intrinsic safety of the system is further reinforced by the application of standard aseptic techniques.

Bioreactors are not required for cultivation, as standard incubators can provide sufficient conditions. Ambient atmospheric $CO_2$ and ordinary LED light sources are adequate for *C. merolae* growth. Together, these features underscore the scalability and practicality of the *Cyanidioschyzon*-based system as a versatile platform for recombinant protein production. Moreover, when cultivated in acidified seawater-based medium, *C. merolae* can be selectively propagated even in harsh outdoor environments (Hirooka et al., 2020; Villegas-Valencia et al., 2023). Thus, from an industrial and economic perspective, this system offers distinct advantages in scalability while minimizing contamination risk.

Another major advantage of the *Cyanidioschyzon*-based system is its ability to produce proteins that are likely to retain native-like structures. This benefit arises from the eukaryotic cellular environment of *C. merolae*, which provides appropriate protein-folding and post-translational modification machinery, including a relatively simple form of N-linked glycosylation (Schulze et al., 2020). Compared with other microalgae, *C. merolae* possesses an advantageously streamlined N-glycosylation system (Liu et al., 2021; Mathieu-Rivet et al., 2014). Many microalgae and plant-based systems can add complex plant-specific sugar residues, such as α1,3-fucose and β1,2-xylose, which might be immunogenic in humans (Bardor et al., 2003). By contrast, *C. merolae* produces proteins with more basic glycan structures that lack these modifications (Schulze et al., 2020). This simplicity can facilitate downstream processing and glyco-engineering.

Given that establishing *C. merolae* transformant lines expressing a target protein typically takes ~1 month, conventional *E. coli*-based expression systems remain far more convenient in terms of the time required for strain construction. However, process optimization might mitigate this disadvantage. For example, the use of a $CO_2$

incubator or mixotrophic cultivation with glycerol has been shown to stimulate respiration and accelerate growth (Moriyama et al., 2015; Villegas-Valencia et al., 2025), which could shorten the cultivation period required for strain establishment. Synonymous codon usage frequencies in protein-coding genes of *C. merolae* have been analyzed, and the impact of codon optimization on gene expression levels in *C. merolae* has been well characterized (Kondo et al., 2024). Leveraging these codon usage data might further enhance recombinant protein accumulation, potentially reducing the overall time required to obtain the target yield of the desired recombinant protein from transformed *C. merolae* cell lines. Further refinement of cultivation equipment and conditions, as well as codon optimization strategies tailored to the intended application and protein of interest, should enhance the overall utility and practicality of the system.

Taken together, these advantages establish the *Cyanidioschyzon*-based system as a highly efficient protein purification platform employing a photosynthetic eukaryote and as an innovative strategy for the production and purification of recombinant proteins.

## MATERIALS AND METHODS
### Cell cultures

The wild-type *Cyanidioschyzon merolae* 10D strain (NIES-3377) and the transformant strains were maintained in 2× Allen's medium (ammonium-based medium) (Allen, 1959) in 60 ml tissue culture flasks (TPP) with shaking at 120 rpm under continuous white light (22 µmol m$^{-2}$ s$^{-1}$) at 42°C. The uracil-auxotrophic M4 strain (Minoda et al., 2004) was maintained in MA2 medium (Ohnuma et al., 2008) supplemented with 0.5 mg/ml uracil and 0.8 mg/ml 5-fluoroorotic acid monohydrate. To prepare cells for protein purification, cultures were diluted to an optical density at 750 nm (OD$_{750}$) of 0.4 in 700 ml of ammonium-based medium and incubated under continuous white light (22 µmol m$^{-2}$ s$^{-1}$) with aeration at 0.6 l of ambient air per min at 42°C for 2 days. Cells were then harvested by centrifugation (2000 **g** for 5 min at room temperature), washed once with PBS, snap-frozen in liquid nitrogen and stored at −80°C until further use.

## Plasmid construction and generation of *C. merolae* strains

The expression vector pHiX was constructed as follows. A functional plasmid backbone was derived from the pGEMTeasy-NIRp-sfGFP plasmid (Fujiwara et al., 2015), which provided essential vector components. The inserted sequence containing the NIR promoter and *sfGFP* coding region was replaced with a 1000-bp upstream region of the *CMK024C* gene, corresponding to the *HiX* promoter. The resulting pHiX plasmid contained the upstream region of *URA5.3* (−2300 to −898), followed by the upstream region of *CMK024C* (−1000 to −1), the downstream region of *TUBB* (+1 to +200), the upstream region of *URA5.3* (−897 to −1), the *URA5.3* open reading frame and the downstream region of *URA5.3* (+1 to +471), in that order.

A DNA fragment encoding His-tagged mVenus for transformation into *C. merolae* cells was generated as follows. The coding sequence for the His tag and mVenus was PCR-amplified from a synthetic DNA template, and the vector backbone was similarly amplified from the pHiX plasmid. These fragments were assembled using Gibson assembly, and the resulting plasmid, pHiX-HiXp-6His-mVenus, was used as a PCR template to generate the final DNA fragment containing the HiXp::6His-mVenus expression cassette along with the *URA5.3* selection marker for transformation of the M4 strain. Polyethylene glycol-mediated transformation of *C. merolae* was performed as described previously (Minoda et al., 2004).

To construct a vector for expressing anti-GFP-VHH fused to FLAG and His tags (pHix-Hixp-αGFP-VHH-FLAG-6His), the coding sequence was PCR-amplified from a synthetic DNA template and assembled with a PCR-amplified pHiX vector backbone via Gibson assembly.

Transformants were screened by colony PCR and fluorescence microscopy. For the strain expressing HiXp::mVenus, the colony PCR-positive rate was 56% (31/55), and 15 transformants that grew robustly in 2-ml liquid culture were selected. For the strain expressing HiXp::anti-GFP-VHH, the colony PCR-positive rate was 73% (30/41), and four of these positive clones expressing putative anti-GFP-VHH were identified by immunoblot analysis. Correct integration at the target locus was confirmed by PCR across the recombination junctions, followed by Sanger sequencing of the amplified products.

All PCRs for plasmid construction were performed using Platinum SuperFi II DNA Polymerase (Thermo Fisher Scientific). DNA fragments were purified with the Wizard SV Gel and PCR Clean-Up System (Promega) and assembled using the NEBuilder HiFi DNA Assembly Cloning Kit (New England Biolabs). Primer sequences generated in this study are listed in Table S1. The plasmid sequence files have been deposited in Zenodo (doi:10.5281/zenodo.18654756).

## Fluorescence microscopy and image processing

Microscopy images were acquired using the same microscope setup and imaging conditions as described previously (Mogi et al., 2025), with the following combination of optical filters applied: excitation filters [490–500HQ (Olympus) for mVenus and FF01-405/10-25 (Semrock) for chloroplasts], custom dichroic mirrors [Di03-R514-t1-25×36 (Semrock) for Venus and T455lp (Chroma) for chloroplasts], and emission filters [FF02-531/22-25 (Semrock) for Venus and FF02-617/73-25 (Semrock) for chloroplasts]. For fluorescence imaging of cells, Z-stacks consisting of 11 images were acquired at 100 nm intervals with a binning of 1 (1×1 pixel size). Background signals were subtracted from the Z-stack images using the rolling-ball background subtraction algorithm (50-pixel radius) in FIJI software (Schindelin et al., 2012). A maximum-intensity Z-projection was then generated from the processed images for presentation. For quantification of fluorescence intensity, the sum-slices projection of the Z-stack images was used. Fluorescence images were binarized using the default automatic thresholding algorithm in FIJI software, after which regions of interest (ROIs) were defined, and fluorescence intensity was measured within the ROIs.

## Quantitative RT-PCR

Total RNA was extracted from *C. merolae* cells as previously described (Miyagishima et al., 2012) with minor modifications. Briefly, cells were lysed in TRIzol reagent (Invitrogen). Following chloroform extraction, total RNA was purified using an RNA Basic Kit (FastGene). During RNA purification, on-column DNase I treatment was performed using the RNase-Free DNase Set (QIAGEN). First-strand cDNA was synthesized from 1 µg of total RNA using ReverTra Ace qPCR RT Master Mix (TOYOBO). RT-qPCR was performed using a Thermal Cycler Dice Real Time System II (Takara) and TB Green Premix Ex Taq II (Takara). Standard curves were constructed using serial dilutions of cDNA and the corresponding primer sets. The cycling conditions were as follows: initial denaturation at 95°C for 30 s, followed by 40 cycles 95°C for 5 s and 60°C for 30 s. Each sample was analyzed in duplicate technical replicates. Data were analyzed using the Thermal Cycler Dice Real Time System software (Takara), and relative transcript levels were calculated using the standard curve method. The *UGDH* gene (CMB031C) encoding UDP-glucose 6-dehydrogenase was selected as the internal reference gene for normalization because RT-qPCR analyses indicated that its expression is highly stable throughout the cell cycle in *C. merolae*. Primers are listed in Table S1.

## Identification of target protein and quantification of expression levels by immunoblotting

Immunoblot analyses for identification of target proteins were performed as follows. *C. merolae* cells were harvested by centrifugation at 2000 $g$ for 5 min and resuspended in SDS sample buffer (FUJIFILM Wako Pure Chemical Corporation). The suspensions were heat-denatured at 95°C for 5 min and centrifuged at 13,000 $g$ for 5 min to remove insoluble material. The resulting supernatants were separated by SDS-PAGE on 12% polyacrylamide gels (resolving gel: 12% acrylamide, 380 mM Tris-HCl pH 8.8, 0.1% SDS, 0.1% APS, 0.05% TEMED; stacking gel: 4.5% acrylamide, 125 mM Tris-HCl pH 6.8, 0.1% SDS, 0.1% APS, 0.1% TEMED) using Precision Plus Protein Dual Color Standards (Bio-Rad) as molecular mass markers, with 20 µg of total protein loaded per lane, and transferred onto PVDF membranes (Immobilon-E, Millipore). Transferred PVDF membranes were blocked overnight at 4°C with Blocking One (Nacalai Tesque) and then incubated with a rabbit anti-GFP primary antibody (ab6556, Abcam) diluted 1:5000 in Blocking One, followed by a horseradish peroxidase-conjugated goat anti-rabbit IgG secondary antibody (ab6721, Abcam) diluted 1:10,000. For the detection of anti-GFP-VHH fused with FLAG and His tags, membranes were incubated with an HRP-conjugated anti-His tag antibody (HRP-66005, Proteintech) diluted 1:5000. An HRP-conjugated anti-DYKDDDDK tag antibody (HRP-66008, Proteintech) diluted 1:5000 was used for the detection of the FLAG tag.

For the calculation of values of the percentage of TSP, standard curves were generated using purified reference proteins. After immunoblotting with 4 µg of total soluble protein for mVenus or 0.5 µg for anti-GFP-VHH, chemiluminescence was developed using the Immobilon Western Chemiluminescent HRP Substrate (Millipore) and signals were detected with a Fusion Solo S imaging system (Vilber). Uncropped images of gel and immunoblot data from this paper are shown in Fig. S2.

## Protein purification from *C. merolae* cells

For protein purification, a total of $5.3×10^{10}$ cells were harvested from 700 ml of culture at a cell density of $7.64×10^7$ cells/ml. Cell numbers were determined using a cell counter (CellDrop BF, DeNovix). The corresponding culture showed an $OD_{750}$ of 0.859. The resulting cell pellet was frozen and resuspended in 10 ml of ice-cold lysis buffer [20 mM Tris-HCl pH 8.0, 150 mM NaCl, 10 mM imidazole, 1.43 mM 2-mercaptoethanol, 10 µg/ml DNase I and cOmplete protease inhibitor (Roche)] by gently pipetting up and down to disrupt cell clumps. The cell lysate was centrifuged at 7000 $g$ for 5 min at 4°C, and the supernatant was collected. The supernatant was then passed through a syringe filter to remove debris. His-tagged proteins were purified using a HisTrap HP column (Cytiva) connected to an NGC Chromatography System (Bio-Rad), with buffer A (20 mM Tris-HCl pH 8.0, 150 mM NaCl) and buffer B (20 mM Tris-HCl pH 8.0, 150 mM NaCl, 500 mM imidazole), according to the manufacturer's instructions. The eluted fractions were dialyzed against dialysis buffer (20 mM Tris-HCl pH 8.0, 150 mM NaCl) to remove imidazole.

## Immunoprecipitation

For immunoprecipitation, cells were lysed as described above, using 2.5 ml of a different lysis buffer [10 mM Tris-HCl pH 7.5, 150 mM NaCl, 0.5 mM EDTA, and cOmplete protease inhibitor (Roche)]. GFP- and FLAG-tagged proteins were purified using anti-GFP or anti-FLAG agarose resins (GFP-Trap or DYKDDDDK Fab-Trap Agarose, ChromoTek) according to the

Journal of Cell Science

manufacturer's protocols. To minimize nonspecific binding to the agarose matrix, Nonidet P40 was added to the clarified supernatant to a final concentration of 0.3%. The supernatant was incubated with equilibrated beads for 1 h at 4°C with gentle rotation. After incubation, the beads were washed repeatedly (six times) with wash buffer [10 mM Tris-HCl pH 7.5, 150 mM NaCl, 0.5 mM EDTA, 0.05% Nonidet P40 Substitute]. For GFP-tagged proteins, bound proteins were eluted by boiling the beads in 2× SDS sample buffer (FUJIFILM Wako Pure Chemical Corporation) for 5 min at 95°C. For FLAG-tagged proteins, bound proteins were eluted with 3× DYKDDDDK peptide (150 µg/ml in PBS, ChromoTek) for 20 min at room temperature, and the eluate was collected by centrifugation at 2500 $g$ for 2 min. Excess 3× DYKDDDDK peptide was removed by buffer exchange using a PD SpinTrap G-25 column (Cytiva) equilibrated with PBS.

## High resolution mass spectrometry
The samples for MS analysis were prepared using the EasyPep Mini MS Sample Prep Kit (Thermo Fisher Scientific) according to the manufacturer's instructions. The resulting peptide samples were dissolved in 15 µl of 0.1% formic acid.

Peptides were separated on an Ultimate3000 RSLCnano (Thermo Fisher Scientific) coupled online to an Orbitrap Exploris 240 mass spectrometer (Thermo Fisher Scientific). The LC separation was performed using a binary solvent system consisting of solvent A (0.1% formic acid in water) and solvent B (0.1% formic acid in acetonitrile). Peptides were loaded onto a C18 analytical column (75 µm×120 mm, 3 µm particle size; Nikkyo Technos) and separated with a gradient starting at 5% B and held for 3 min, followed by a linear increase from of 5 to 45% solvent B over 93 min at a flow rate of 300 nl/min. The electrospray voltage was set to 2.0 kV, and the ion transfer tube temperature was maintained at 250°C.

The mass spectrometer was operated in data-dependent acquisition mode. Full MS scans were acquired in the Orbitrap over a mass range of $m/z$ 350–1500 at a resolution of 60,000, followed by MS/MS scans using higher-energy collisional dissociation (HCD) with a normalized collision energy of 30%.

Raw mass spectrometry data were processed using Proteome Discoverer software (version 2.5, Thermo Fisher Scientific). Spectra were searched against a database containing mVenus and *C. merolae* protein sequences (Matsuzaki et al., 2004) using the Sequest HT search engine. The following parameters were used: enzyme specificity was set to trypsin with up to two missed cleavages; precursor mass tolerance was set to 10 ppm and fragment mass tolerance to 0.02 Da. Carbamidomethylation of cysteine was set as a fixed modification, whereas oxidation of methionine and loss of the initiator methionine and/or acetylation of the protein N-terminus were set as variable modifications. Peptide-spectrum matches (PSMs) were validated using the Percolator algorithm, and a false discovery rate (FDR) of 1% was applied at both the peptide and protein levels, with a minimum peptide length of six amino acids.

## Acknowledgements
We thank our lab colleagues for their support and advice during this project.

## Competing interests
The authors declare no competing or financial interests.

## Author contributions
Conceptualization: Y.M., Y.Y.; Funding acquisition: F.Y., Y.Y.; Investigation: Y.M., S.T., S.N., S.G., F.Y., Y.Y.; Methodology: Y.M., Y.Y.; Project administration: Y.Y.; Supervision: Y.Y.; Visualization: Y.M., Y.Y.; Writing – original draft: Y.M., Y.Y.; Writing – review & editing: Y.M., S.T., S.N., S.G., F.Y., Y.Y.

## Funding
This work was supported by FOREST from the Japan Science and Technology Agency (JPMJFR2316 to Y.Y.) and Japan Society for the Promotion of Science KAKENHI (nos. JP18K06325, 22H02653 to Y.Y. and no. 25K09727 to F.Y.). Open Access funding provided by University of Tokyo. Deposited in PMC for immediate release.

## Data and resource availability
Plasmid sequence files have been deposited in Zenodo (doi:10.5281/zenodo.18654756). All other relevant data and details of resources can be found within the article and its supplementary information.

## Peer review history
The peer review history is available online at https://journals.biologists.com/jcs/lookup/doi/10.1242/jcs.264207.reviewer-comments.pdf

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
