## [Peer Review File · Journal of Cell Science]

A high-yield protein expression platform in the unicellular red alga *Cyanidioschyzon merolae*

Yuko Mogi, Shogo Tsushima, Shotaro Nagai, Shinichi Gima, Fumi Yagisawa and Yamato Yoshida

DOI: 10.1242/jcs.264207

Editor: Charlotte Kirchhelle

Review timeline

Original submission:	11 June 2025
Editorial decision:	30 June 2025
First revision received:	11 November 2025
Editorial decision:	11 December 2025
Second revision received:	20 December 2025
Editorial decision:	19 January 2026
Third revision received:	20 January 2026
Accepted:	22 January 2026

Original submission

First decision letter

MS ID#: jcs.264207

MS TITLE: CZON-pure: An efficient platform for high-yield protein isolation using a unicellular alga

AUTHORS: Yuko Mogi; Shogo Tsushima; Shotaro Nagai; Yamato Yoshida

ARTICLE TYPE: Tools and Resources

Dear Dr Yoshida,

We have now reached a decision on the above manuscript.

To see the reviewers' reports and a copy of this decision letter, please go to:

As you will see, the reviewers raise a number of serious and substantial criticisms that prevent me from accepting the paper at this stage. I would like to particularly emphasize that the manuscript should adhere to accepted standards in the field for the reporting of yield efficiency, and technical ambiguities/limitations should be addressed experimentally as well as discussed. Despite the current shortcomings of the manuscript, the reviewers suggest, that a revised version might prove acceptable, if you can address their concerns. If you think that you can deal satisfactorily with the criticisms on revision, I would be pleased to see a revised manuscript. We would then return it to the reviewers.

Reviewer 1

SUMMARY OF THE ADVANCE MADE IN THIS PAPER AND ITS POTENTIAL SIGNIFICANCE TO THE FIELD

The authors describe the identification of a new promoter based on the upstream region of the "HiX" gene of the CMK024C locus. They then describe its use to express the YFP gene (mVenus) via

HR transformation and complementation of the URA locus for selection in the M4 background strain.

The authors claim the target protein is accumulated up to 3% of the "of the total cellular proteins" being "one of the most abundant bands" in SDS AGE followed by Coomassie staining. After following His-tagged purification, they claim 1 mg purified mVenus protein from 1 g wet mass cells. The authors then describe that another promoter, NiR is used to yield 68% of the amount of the HiX promoter yielding 0.4 mg mVenus from 1 gram wet cell mass.

So this manuscript, if results are accurate, could be a very valuable addition to the body of knowledge around *C. merolae* and its application in biotechnology.

SUGGESTIONS TO AUTHORS

However, there are some issues which need to be addressed before it can be taken seriously for publication.

For better or worse, algal biotechnology relies on some metrics of measurement to compare across cell systems and between mutants. Two metrics of importance are: total soluble protein (% TSP) and dry mass of cells as a benchmark for growth and system yields.

The authors provide neither of these units. How much cell mass is made per unit time in a cultivation? Obviously, the system shown is not an optimized bioreactor, so I expect ~1 g dry biomass per L culture. 1 g wet weight of cell pellet is not a comparable unit of mass which can be easily replicated by others. Please make units comparable to gram dry mass / L culture.

The authors provide information for protein expression from only a single transformant with the HiX promoter. As we know that multiple transgene insertions can occur into the genome of *C. merolae*, in addition to HR at the target locus - it is possible that this is not a promoter induced high protein expression, but rather a multi-copy insertion transformant that is better than others in a population. Quantification of protein expression amounts as % total soluble protein against a purified mVenus protein standard is required in at least three separate transformants to make calculations of expression activity. And each of these require genotyping with both PCR at the target locus and copy-number assessment through Southern blotting, quick genome sequencing, or qPCR to make a claim of strong promoter activity.

Band shown on SDS page may be an artefact of cultivation conditions, as the upper portion of this lane is also different from the WT, again here, more than a single transformant is required to be shown to make these claims. Given the signal on Western blot, it is possible this indicated band in SDS PAGE is not the mVenus protein, as a much stronger signal would be observed. This could be shown with in gel fluorescence assessment of the mVenus signal as is now possible in many gel doc systems. A dilution of mVenus purified protein standard at known quantities loaded next to known amounts of soluble protein cell extract will enable quantification.

It is concerning to make a claim that the NiR promoter is achieving 68% of the HiX promoter as this amount would imply >1% total protein in the cell is mVenus, when the NiR has been previously used without high expression claims.

A 2-hourly sampling though a 24 hour cycle and a 2-sample per day - 5 day growth curve PAGE and Western blot would be beneficial to show activity throughout the growth curve for both promoters. I would like to see expression of other proteins with the HiX promoter - just mVenus is fine, but functionally not important - how does it work with other soluble proteins, like maybe an antifreeze protein, alternative sweetener protein, or nanobody?

It is further concerning that no genomic location information is provided about the NiR promoter. Is it CMG019C from Fujiwara et al., 2015? It feels like it was added to the claims as an afterthought. As this is not commonly used in other publications - much more information is required about its use - or it should be omitted entirely from the manuscript. Does the promoter activity last for several hours after medium switch? How long is its high activity noted for? Is it turned on in the presence of nitrate or the absence of ammonium which causes its activity?

All plasmids should be exported to .gb format and combined in a single .txt. file. An example of this is: Appendix A. Supplementary data File 1 of this paper: Upon downloading the supplement, the file is named "1-s2.0-S2214030123000044-mmc1"

<https://www.sciencedirect.com/science/article/pii/S2214030123000044?via=ihub#appsec1>

The abbreviation CZON is problematic and not helpful. Although I understand the desire to make a catchy slang - is not following scientific norms nor is it particularly helpful - *C. merolae* is an appropriate abbreviation of *Cyanidioschyzon merolae* (the first name being hard to say). 'Cm' would be the only consistent and other appropriate abbreviation. Yoshida Sensei has used this 'CZON' in one previous publication from 2021 - however it is not adopted by the community and I would not

encourage its further use.

If the authors can deal with these issues - they can have a nice publication of value to the community, however, I think their protein expression claims need further validation before being taken seriously.

Reviewer 2

SUMMARY OF THE ADVANCE MADE IN THIS PAPER AND ITS POTENTIAL SIGNIFICANCE TO THE FIELD

Mogi and her/his colleagues established the CZON-pure protein expression system using unicellular red algae.

In this system, protein expression is induced by a promoter responsive to nitrogen sources to handle proteins that are toxic to the cells.

While the data are clear, several revisions are needed, as outlined in the comments below.

SUGGESTIONS TO AUTHORS

1) They claim that the production cost of CZON-pure is lower compared to other expression systems. However, there is scant evidence to support this claim. It is necessary to develop the discussion based on concrete evidence.

2) In CZON-pure, it is necessary to knock in the target gene into the genome. It is assumed that obtaining such transformants will require a longer period than with *E. coli*. There should be a discussion of such drawbacks (future challenges) in addition to the claimed advantages.

3) A notable advantage of CZON-pure is the ability to synthesize proteins that closely resemble their native structures. This is something that cannot be achieved with *E. coli*, and it would be beneficial to describe and discuss this point. If possible, please provide comparative data on proteins synthesized by CZON-pure and conventional methods.

4) The information provided in the figure legends is insufficient and unhelpful for the readers. It is necessary to describe the required information thoroughly and carefully.

5) The induction system using the NIR promoter in *C. merolae* has been previously published by several groups (for example, <https://doi.org/10.3389/fpls.2015.00657>, <https://doi.org/10.1093/plphys/kiaf106>). It is necessary to cite these references and clarify what the novelty of your study is.

First revision

Author response to reviewers' comments

Responses to Reviewers

We sincerely thank the reviewers for their careful evaluation of our manuscript and for the constructive comments and suggestions. We have carefully considered all points raised and revised the manuscript accordingly. Below, we provide a point-by-point response to each comment.

Please note that reviewer comments are shown in **blue**, and our responses are shown in **black**.

Comments from the Reviewers:

Reviewer 1: SUMMARY OF THE ADVANCE MADE IN THIS PAPER AND ITS POTENTIAL SIGNIFICANCE TO THE FIELD

The authors describe the identification of a new promoter based on the upstream region of the "HiX" gene of the CMK024C locus. They then describe its use to express the YFP gene (mVenus) via HR transformation and complementation of the URA locus for selection in the M4 background strain.

The authors claim the target protein is accumulated up to 3% of the "of the total cellular proteins" being "one of the most abundant bands" in SDS AGE followed by Coomassie staining. After following His-tagged purification, they claim 1 mg purified mVenus protein from 1 g wet mass cells. The authors then describe that another promoter, NiR is used to yield 68% of the amount of the HiX promoter yielding 0.4 mg mVenus from 1 gram wet cell mass.

So this manuscript, if results are accurate, could be a very valuable addition to the body of knowledge around *C. merolae* and its application in biotechnology.

- We sincerely thank Reviewer 1 for the positive and encouraging summary of our study. We appreciate the recognition of our efforts to identify a novel promoter in *C. merolae* and to demonstrate its potential application for heterologous protein expression. We are grateful for the reviewer's acknowledgment that our findings could contribute to advancing the biotechnological use of *C. merolae*. We hope that this promoter system will serve as a useful tool for future studies of gene regulation and recombinant protein production in red algae.

SUGGESTIONS TO AUTHORS

However, there are some issues which need to be addressed before it can be taken seriously for publication.

For better or worse, algal biotechnology relies on some metrics of measurement to compare across cell systems and between mutants. Two metrics of importance are: total soluble protein (% TSP) and dry mass of cells as a benchmark for growth and system yields.

The authors provide neither of these units. How much cell mass is made per unit time in a cultivation? Obviously, the system shown is not an optimized bioreactor, so I expect ~1 g dry biomass per L culture. 1 g wet weight of cell pellet is not a comparable unit of mass which can be easily replicated by others. Please make units comparable to gram dry mass / L culture.

The authors provide information for protein expression from only a single transformant with the HiX promoter. As we know that multiple transgene insertions can occur into the genome of *C. merolae*, in addition to HR at the target locus - it is possible that this is not a promoter induced high protein expression, but rather a multi-copy insertion transformant that is better than others in a population.

Quantification of protein expression amounts as % total soluble protein against a purified mVenus protein standard is required in at least three separate transformants to make calculations of expression activity. And each of these require genotyping with both PCR at the target locus and copy-number assessment through Southern blotting, quick genome sequencing, or qPCR to make a claim of strong promoter activity.

- Thank you for these important comments. We have now measured the percentage of total soluble protein (%TSP) in three independent transformant strains expressing the mVenus protein. The quantified %TSP values are 1.27, 1.46, and 3.20. These data are now presented in Figure 3 and described in the main text.
- We also characterized the fluorescence intensity of mVenus expression in individual cells across 15 independent transformant strains (Figure 2D). Statistical analysis using the Mann–Whitney test showed that the strains could be classified into four groups based on fluorescence intensity. The ratio of fluorescence intensity between the highest-expressing strain (#52) and the lowest-expressing strain (#23) was approximately 7.0, suggesting that multi-copy insertions and/or other unpredictable effects related to expression activity may have occurred in some strains, as you pointed out. Among these, four strains in group *c* exhibited intermediate fluorescence levels between #23 and #52, with approximately twofold higher mVenus RNA expression compared with #23, as determined by qPCR (Figure 2F).
- Given the purpose of target protein purification, establishing strains showing moderate expression levels appears to be the most practical approach, based on the observed distribution of fluorescence intensities across the 15 transformants. Our results indicate that higher-expression strains can be readily obtained using the *C. merolae* transformation method. As immunoblot analysis confirmed that strain #1 expresses an mVenus protein of the correct molecular mass, we therefore selected strain #1, which shows a moderate expression level, for subsequent analyses.

Band shown on SDS page may be an artefact of cultivation conditions, as the upper portion of this lane is also different from the WT, again here, more than a single transformant is required to be shown to make these claims. Given the signal on Western blot, it is possible this indicated band in SDS PAGE is not the mVenus protein, as a much stronger signal would be observed. This could be shown with in gel fluorescence assessment of the mVenus signal as is now possible in many gel doc systems. A dilution of mVenus purified protein standard at known quantities loaded next to known amounts of soluble protein cell extract will enable quantification.

- We have removed the arrowhead from the gel image in Figure 3A to avoid any misunderstanding that a specific CBB-stained band corresponds to mVenus.

It is concerning to make a claim that the NiR promoter is achieving 68% of the HiX promoter as this amount would imply >1% total protein in the cell is mVenus, when the NiR has been previously used without high expression claims.

A 2-hourly sampling though a 24 hour cycle and a 2-sample per day - 5 day growth curve PAGE and Western blot would be beneficial to show activity throughout the growth curve for both

promoters.

I would like to see expression of other proteins with the HiX promoter - just mVenus is fine, but functionally not important - how does it work with other soluble proteins, like maybe an antifreeze protein, alternative sweetener protein, or nanobody?

- Instead of providing additional results for the NiR promoter, we examined the production and purification of an anti-GFP nanobody in *C. merolae* cells (Figure 5C-E). This experiment was conducted to demonstrate the versatility of the HiX promoter for expressing different soluble proteins. The expression level of the FLAG-tagged anti-GFP nanobody was approximately 0.34% TSP. We confirmed that the purified anti-GFP nanobody obtained through immunoprecipitation successfully recognized mVenus in immunoblotting analysis.
- These results demonstrate that the HiX promoter can effectively drive the expression of other functional and soluble proteins beyond mVenus.

It is further concerning that no genomic location information is provided about the NiR promoter. Is it CMG019C from Fujiwara et al., 2015? It feels like it was added to the claims as an afterthought. As this is not commonly used in other publications - much more information is required about its use - or it should be omitted entirely from the manuscript. Does the promoter activity last for several hours after medium switch? How long is its high activity noted for? Is it turned on in the presence of nitrate or the absence of ammonium which causes its activity?

- We have removed all data related to the NiR promoter from the manuscript and replaced them with another example of protein expression in *C. merolae*, demonstrating nanobody production.
- In addition, we added results of immunoprecipitation combined with proteomic analysis using the mVenus protein (Figure 5A and B). The high specificity of the purified mVenus signals identified by mass spectrometry indicates that mVenus-tagged proteins and their cofactors can be effectively distinguished from background proteins, providing a valuable approach for investigating molecular mechanisms in this organism. This revision improves the focus of the manuscript on the established expression system using the HiX promoter.

All plasmids should be exported to .gb format and combined in a single .txt. file. An example of this is: Appendix A. Supplementary data File 1 of this paper: Upon downloading the supplement, the file is named "1-s2.0-S2214030123000044-mm1" <https://www.sciencedirect.com/science/article/pii/S2214030123000044?via=ihub#appsec1>

The abbreviation CZON is problematic and not helpful. Although I understand the desire to make a catchy slang - is not following scientific norms nor is it particularly helpful - *C. merolae* is an appropriate abbreviation of *Cyanidioschyzon merolae* (the first name being hard to say).

'Cm' would be the only consistent and other appropriate abbreviation. Yoshida Sensei has used this 'CZON' in one previous publication from 2021 - however it is not adopted by the community and I would not encourage its further use.

- In accordance with the reviewer's suggestions, we have prepared a plasmid dataset in GenBank (.gb) format and combined all plasmid entries into a single text file. We have also revised the manuscript title and removed the term "CZON" to maintain consistency with scientific conventions.

If the authors can deal with these issues - they can have a nice publication of value to the community, however, I think their protein expression claims need further validation before being taken seriously.

Reviewer 2: SUMMARY OF THE ADVANCE MADE IN THIS PAPER AND ITS POTENTIAL SIGNIFICANCE TO THE FIELD

Mogi and her/his colleagues established the CZON-pure protein expression system using unicellular red algae.

In this system, protein expression is induced by a promoter responsive to nitrogen sources to handle proteins that are toxic to the cells.

While the data are clear, several revisions are needed, as outlined in the comments below.

- We sincerely thank Reviewer 2 for the constructive summary and thoughtful evaluation of our work. We greatly appreciate the recognition of our efforts to establish the *C. merolae*-based protein expression system and its potential for controlled expression of proteins that are toxic to cells. We also appreciate the reviewer's helpful comments and suggestions for improvement, which have guided us in revising the manuscript.

- We have carefully addressed all points raised in the following responses.

SUGGESTIONS TO AUTHORS

1) They claim that the production cost of CZON-pure is lower compared to other expression systems. However, there is scant evidence to support this claim. It is necessary to develop the discussion based on concrete evidence.

- We agree with the reviewer that the cost advantage of the CZON-pure system should not be overstated without quantitative data. We have therefore modified the statement to describe this aspect more cautiously, noting that the system has the potential to reduce production costs because it requires only inorganic nutrients and light energy for cultivation.
- We have revised the Discussion section accordingly.

2) In CZON-pure, it is necessary to knock in the target gene into the genome. It is assumed that obtaining such transformants will require a longer period than with *E. coli*. There should be a

discussion of such drawbacks (future challenges) in addition to the claimed advantages.

- We appreciate the reviewer's valuable comment. We agree that the requirement for genomic integration of the target gene is a current limitation of the CZON-pure system compared to *E. coli*, as obtaining stable transformants requires additional time. We have now added a short discussion of this point in the revised manuscript.
- We also note that, once a transformant is established, the expression cassette remains stably inherited without plasmid loss or the need for antibiotics, which can be an advantage for long-term protein production. Future efforts will focus on developing more rapid transformation and screening methods to further improve the efficiency of the system.

3) A notable advantage of CZON-pure is the ability to synthesize proteins that closely resemble their native structures. This is something that cannot be achieved with *E. coli*, and it would be beneficial to describe and discuss this point. If possible, please provide comparative data on proteins synthesized by CZON-pure and conventional methods.

- We appreciate the reviewer's insightful comment. We agree that one of the major advantages of the CZON-pure system is its ability to produce proteins that are likely to retain native-like structures, owing to the eukaryotic cellular environment of *C. merolae*, which provides proper folding and post-translational modification machinery.
- We have now added a short paragraph in the Discussion describing this potential advantage.
- Although we have not yet conducted direct comparative analyses between CZON-pure-derived proteins and those produced in *E. coli*, such studies are planned as part of our future work.

4) The information provided in the figure legends is insufficient and unhelpful for the readers. It is necessary to describe the required information thoroughly and carefully.

- We appreciate the reviewer's comment. We have thoroughly revised all figure legends to make them self-contained and informative. Each legend now includes: sample size (cells/biological replicates), experimental conditions, strain/genotype and construct details (promoter, tag, integration locus), exact units and scales, image acquisition parameters (microscope, scale bars), loading amounts and molecular-weight markers for gels/blots, antibody information and dilutions, quantitative metrics (mean \pm s.d.), the definition of error bars, and statistical tests. Abbreviations and color schemes are defined at first appearance, and cross-references to the corresponding Methods and Supplementary Tables are provided.

5) The induction system using the NIR promoter in *C. merolae* has been previously published by several groups (for example, <https://doi.org/10.3389/fpls.2015.00657>, <https://doi.org/10.1093/plphys/kiaf106>). It is necessary to cite these references and clarify what the

novelty of your study is.

- We thank the reviewer for this important comment. We have removed all data related to the NiR promoter from the manuscript and replaced them with another example of protein expression in *C. merolae*, demonstrating nanobody production.
- This revision improves the focus of the manuscript on the established expression system using the HiX promoter.

Second decision letter

MS ID#: jcs.264207R1

MS TITLE: A high-yield protein expression platform in the unicellular red alga *Cyanidioschyzon merolae*

AUTHORS: Yuko Mogi; Shogo Tsushima; Shotaro Nagai; Yamato Yoshida; Fumi Yagisawa; Shinichi Gima

ARTICLE TYPE: Tools and Resources

Dear Dr Yoshida,

We have now reached a decision on the above manuscript.

As you will see, the reviewers gave favourable reports but raised some critical points that will require amendments to your manuscript. I invite you to carefully address all of the remaining reviewer's comments. I am looking forward to receiving your revised manuscript as I would like to be able to accept your paper, if you are able to adequately address the remaining comments.

Please upload both a 'clean' version of your Word file, along with a highlighted version clearly showing where you have made changes in the revised manuscript. Please avoid using 'Track changes' in Word files as these are lost in PDF conversion. As the changes in the current version were not obvious, I would appreciate if you could highlight both original revisions and second revisions.

I should be grateful if you would also provide a point-by-point response detailing how you have dealt with the points raised by the reviewers in the 'Response to Reviewers' box. Please attend to all of the reviewers' comments. If you do not agree with any of their criticisms or suggestions please explain clearly why this is so.

Reviewer 1

SUMMARY OF THE ADVANCE MADE IN THIS PAPER AND ITS POTENTIAL SIGNIFICANCE TO THE FIELD

The manuscript presents the development of a recombinant protein expression and purification platform using the unicellular red alga *Cyanidioschyzon merolae*. A novel promoter (HiX), identified from the upstream region of the CMK024C locus through transcriptome and translatoome analyses, was shown to drive the highest constitutive expression across the cell cycle. Functional validation demonstrated stable transgene expression under HiX control. An expression vector (pHiX) was engineered for homologous recombination into a genomic safe harbor locus, enabling stable integration and high-level expression of target proteins. Using mVenus as a model, expression levels reached up to 3.2% of total soluble protein (TSP) in high-expressing strains, with moderate-expression strains (~1.3-1.5% TSP) selected for downstream applications. Expression was consistent throughout the cell cycle, and transformants exhibited normal growth phenotypes. The absence of a cell wall in *C. merolae* facilitated efficient protein release via a single freeze-thaw cycle, and

His-tagged mVenus was purified at yields of approximately 1 mg per gram of wet cell mass using standard IMAC. Additionally, an anti-GFP nanobody was expressed and purified (~0.34% TSP), and immunoprecipitation coupled with mass spectrometry demonstrated the system's utility for isolating protein complexes with high specificity. The platform leverages the organism's ability to grow under highly acidic conditions using only inorganic nutrients, reducing contamination risk and potentially lowering costs. While offering scalability and eukaryotic folding/post-translational modification advantages, limitations include the time required for genomic integration and the need for further optimization for industrial-scale production. Data supporting cost-effectiveness and comparative performance versus conventional systems remain qualitative. The authors have greatly improved the manuscript since the original submitted version. There are minor things to be corrected before it can be further considered.

SUGGESTIONS TO AUTHORS

Although use of 'total soluble protein' as a measurement has been presented, One section uses g wet cell mass still (Streamlined Purification of....). This has no units of cell number or OD harvested to make this cell mass - it may be something not important to that authors, because it is an approximation of a cell pellet from dense culture, but it is important for being precise in the paper. What is the dry biomass weight? This is especially important because the next section is in mg/ g total soluble protein.

On page 9, it is state 'immunoprecipitation and proteomic analyses...' but I do not think this is the right term. Protein assessment by molecular biology techniques?

The last paragraph pf "Selective protein purification" section is Discussion and should be moved or removed.

Overall, the manuscript is close to acceptable, but requires the minor considerations.

Reviewer 2

Comments to Author:

SUMMARY OF THE ADVANCE MADE IN THIS PAPER AND ITS POTENTIAL SIGNIFICANCE TO THE FIELD
Mogi and her/his colleagues established the CZON-pure protein expression system using unicellular red algae.

In this system, protein expression is induced by a promoter responsive to nitrogen sources to handle proteins that are toxic to the cells.

While the data are clear, several revisions are needed, as outlined in the comments below.

SUGGESTIONS TO AUTHORS

1) They claim that the production cost of CZON-pure is lower compared to other expression systems. However, there is scant evidence to support this claim. It is necessary to develop the discussion based on concrete evidence.

2) In CZON-pure, it is necessary to knock in the target gene into the genome. It is assumed that obtaining such transformants will require a longer period than with E. coli. There should be a discussion of such drawbacks (future challenges) in addition to the claimed advantages.

3) A notable advantage of CZON-pure is the ability to synthesize proteins that closely resemble their native structures. This is something that cannot be achieved with E. coli, and it would be beneficial to describe and discuss this point. If possible, please provide comparative data on proteins synthesized by CZON-pure and conventional methods.

4) The information provided in the figure legends is insufficient and unhelpful for the readers. It is necessary to describe the required information thoroughly and carefully.

5) The induction system using the NIR promoter in *C. merolae* has been previously published by several groups (for example, <https://doi.org/10.3389/fpls.2015.00657>,

<https://doi.org/10.1093/plphys/kiaf106>). It is necessary to cite these references and clarify what the novelty of your study is.

Second revision

Author response to reviewers' comments

We greatly appreciate the reviewers' additional review of our revised manuscript and their helpful follow-up comments. We have addressed all remaining concerns and further revised the manuscript to improve clarity and rigor. Below, we provide a detailed point-by-point response to each comment.

Please note that reviewer comments are shown in **blue**, and our responses are shown in **black**. In the manuscript, revised sentences are highlighted in **yellow** for the first-round revision and in **blue** for the second-round revision.

Reviewer 1: SUMMARY OF THE ADVANCE MADE IN THIS PAPER AND ITS POTENTIAL SIGNIFICANCE TO THE FIELD

The manuscript presents the development of a recombinant protein expression and purification platform using the unicellular red alga *Cyanidioschyzon merolae*. A novel promoter (HiX), identified from the upstream region of the *CMK024C* locus through transcriptome and translome analyses, was shown to drive the highest constitutive expression across the cell cycle. Functional validation demonstrated stable transgene expression under HiX control. An expression vector (pHiX) was engineered for homologous recombination into a genomic safe harbor locus, enabling stable integration and high-level expression of target proteins. Using mVenus as a model, expression levels reached up to 3.2% of total soluble protein (TSP) in high-expressing strains, with moderate-expression strains (~1.3-1.5% TSP) selected for downstream applications. Expression was consistent throughout the cell cycle, and transformants exhibited normal growth phenotypes. The absence of a cell wall in *C. merolae* facilitated efficient protein release via a single freeze-thaw cycle, and His-tagged mVenus was purified at yields of approximately 1 mg per gram of wet cell mass using standard IMAC. Additionally, an anti-GFP nanobody was expressed and purified (~0.34% TSP), and immunoprecipitation coupled with mass spectrometry demonstrated the system's utility for isolating protein complexes with high specificity. The platform leverages the organism's ability to grow under highly acidic conditions using only inorganic nutrients, reducing contamination risk and potentially lowering costs. While offering scalability and eukaryotic folding/post-translational modification advantages, limitations include the time required for genomic integration and the need for further optimization for industrial-scale production. Data supporting cost-effectiveness and comparative performance versus conventional systems remain qualitative. The authors have greatly improved the manuscript since the original submitted version. There are minor things to be corrected before it can be further considered.

We sincerely appreciate Reviewer 1's thorough second-round review and the thoughtful summary highlighting the technical advances, strengths, and potential impact of our platform. We are encouraged by the positive evaluation and thank the reviewer for recognizing the improvements made since the initial submission. All remaining minor comments have been carefully addressed in the revised manuscript.

SUGGESTIONS TO AUTHORS

>Although use of 'total soluble protein' as a measurement has been presented, One section uses g wet cell mass still (Streamlined Purification of....). This has no units of cell number or OD harvested to make this cell mass - it may be something not important to that authors, because it is an approximation of a cell pellet from dense culture, but it is important for being precise in the paper. What is the dry biomass weight? This is especially important because the next section is

in mg/ g total soluble protein.

Thank you for pointing this out. We have revised the sentence referring to “gram wet cell mass” and replaced it with more accurate quantitative descriptors, such as the amount of total soluble protein and the total number of cells used in the experiment. Cell numbers were primarily determined using an automated cell counter (CellDrop BF, DeNovix), and the optical density at 750 nm (OD750) was also provided for reference. In addition, we replaced “gram wet cell mass” with appropriate terms throughout the manuscript.

Please see page 2, lines 12-14 (Abstract); page 4, lines 29-30 (Introduction); page 8, lines 25-27 (Results); and page 17, lines 13-15 (Materials and Methods).

=

>On page 9, it is state 'immunoprecipitation and proteomic analyses...' but I do not think this is the right term. Protein assessment by molecular biology techniques?

We thank you for your comment and agree that the sentence required revision. We have rewritten the sentence as follows: “We next performed immunoprecipitation-based purification using mVenus expressed in *C. merolae* and evaluated the enrichment specificity of the workflow”. We believe that this revision more appropriately describes the analytical process and clearly refers to the corresponding section.

Please see page 9, lines 10-11.

=

>The last paragraph pf "Selective protein purification" section is Discussion and should be moved or removed. Overall, the manuscript is close to acceptable, but requires the minor considerations.

We agree that the section requires revision. The indicated text has been deleted from the last paragraph, and we have expanded the Discussion to better describe the advantages of our system and to compare it with other algal-based purification systems, citing the relevant pioneering studies. We also discuss possibilities for further optimization and engineering developments, as well as future perspectives.

Please see page 9, line 31; page 10, lines 1-11.

Reviewer 2: Thank you for revising the manuscript.

>My previous comments 1-3 have been addressed mainly through revisions in the discussion section. However, the resulting discussion remains rather superficial. This is further supported by the fact that no references are cited. I believe it is necessary to discuss the significance of this study in greater depth.

We sincerely thank you for your second-round review and for your continued constructive feedback. We agree with your assessment that the Discussion section required substantial revision. Accordingly, we have comprehensively rewritten the Discussion and added new text to more clearly articulate the advantages of our system and to compare it with other algal-based purification platforms, citing the relevant pioneering studies. We believe that these revisions have strengthened the Discussion and more clearly delineated the scientific significance and positioning of our system within the broader landscape of protein purification platforms.

Please see page 11, lines 22-31; page 12, lines 1-8, 16-21, and 24-31; page 13, lines 1-2 and 6-17.

>Although this is not a critical issue, I recommend clearly indicating the sections where changes have been made in the response letter. This will make it clear which parts the authors are referring to.

In accordance with your suggestion, we have indicated the revised sections and the specific sentences, together with the corresponding page and line numbers, for each comment.

Third decision letter

MS ID#: jcs.264207R2

MS TITLE: A high-yield protein expression platform in the unicellular red alga *Cyanidioschyzon merolae*

AUTHORS: Yuko Mogi; Shogo Tsushima; Shotaro Nagai; Shinichi Gima; Fumi Yagisawa; Yamato Yoshida

ARTICLE TYPE: Tools and Resources

Dear Dr Yoshida,

We have now reached a decision on the above manuscript.

As you will see, the reviewers (and myself) were content that you addressed all of their concerns with your thorough revision of the manuscript. There is only one minor concern regarding the wording in the discussion, which can be addressed very easily. I would be very grateful if you could do this. As the final revision is very minor, I would like to let you know that I will evaluate this myself and the manuscript would not be sent back out to reviewers before acceptance.

Reviewer 1

Comments to Author:

Paper is suitable for publication.

Reviewer 2

Comments to Author:

Thank you for addressing my concerns.

I believe the manuscript has improved considerably overall.

If I may point out one specific area for further refinement, the following statement in the Discussion section may be somewhat overstated:

"The most efficient" should be moderated in tone, as it may overstate the current findings.

Third revision

Author response to reviewers' comments

We greatly appreciate the reviewers' additional evaluation of our revised manuscript and their thoughtful comments. We have addressed the point raised and revised the manuscript to improve clarity and accuracy. Below, we provide a detailed response to the comment.

Please note that reviewer comments are shown in **blue**, and our responses are shown in **black**. In the manuscript, revised sentences are highlighted in **yellow** for the first-round revision, in **blue** for the second-round revision, and in **green** for the third-round revision.

Comments from the Reviewers:

Reviewer 1: Paper is suitable for publication.

Reviewer 2: Thank you for addressing my concerns.

>I believe the manuscript has improved considerably overall.

>If I may point out one specific area for further refinement, the following statement in the Discussion section may be somewhat overstated:

>"The most efficient" should be moderated in tone, as it may overstate the current findings.

We thank you for this comment and agree that the sentence required moderation. We have rewritten it as follows: "Taken together, these advantages establish the *Cyanidioschyzon*-based system as a **highly efficient** protein purification platform employing a photosynthetic eukaryote and as an innovative strategy for the production and purification of recombinant proteins." We believe that this revision more appropriately reflects our findings.

We also made the following two minor revisions to improve readability and the accuracy of the methodological description.

In Figure 1A, labels for RNA read counts and ribosome footprint read counts are shown in red and blue, respectively.

In the Materials and Methods section, the buffer composition used for immunoprecipitation was revised.

Fourth decision letter

MS ID#: jcs.264207R3

MS Title: A high-yield protein expression platform in the unicellular red alga *Cyanidioschyzon merolae*

Authors: Yuko Mogi; Shogo Tsushima; Shotaro Nagai; Shinichi Gima; Fumi Yagisawa; Yamato Yoshida
Article Type: Tools and Resources

Dear Dr Yoshida,

Thank you for completing your final revision. I am happy to tell you that your manuscript has been accepted for publication in Journal of Cell Science, pending standard publication integrity checks.